# Gene Expression Profiling as a Novel Diagnostic Tool for Neurodegenerative Disorders

**DOI:** 10.3390/ijms24065746

**Published:** 2023-03-17

**Authors:** Olaia Martínez-Iglesias, Vinogran Naidoo, Juan Carlos Carril, Silvia Seoane, Natalia Cacabelos, Ramón Cacabelos

**Affiliations:** EuroEspes Biomedical Research Center, International Center of Neuroscience and Genomic Medicine, 15165 Bergondo, Corunna, Spain

**Keywords:** neurodegenerative disorders, diagnostic biomarker, gene expression, APOE

## Abstract

There is a lack of effective diagnostic biomarkers for neurodegenerative disorders (NDDs). Here, we established gene expression profiles for diagnosing Alzheimer’s disease (AD), Parkinson’s disease (PD), and vascular (VaD)/mixed dementia. Patients with AD had decreased APOE, PSEN1, and ABCA7 mRNA expression. Subjects with VaD/mixed dementia had 98% higher PICALM mRNA levels, but 75% lower ABCA7 mRNA expression than healthy individuals. Patients with PD and PD-related disorders showed increased SNCA mRNA levels. There were no differences in mRNA expression for OPRK1, NTRK2, and LRRK2 between healthy subjects and NDD patients. APOE mRNA expression had high diagnostic accuracy for AD, and moderate accuracy for PD and VaD/mixed dementia. PSEN1 mRNA expression showed promising accuracy for AD. PICALM mRNA expression was less accurate as a biomarker for AD. ABCA7 and SNCA mRNA expression showed high-to-excellent diagnostic accuracy for AD and PD, and moderate-to-high accuracy for VaD/mixed dementia. The APOE E4 allele reduced APOE expression in patients with different APOE genotypes. There was no association between PSEN1, PICALM, ABCA7, and SNCA gene polymorphisms and expression. Our study suggests that gene expression analysis has diagnostic value for NDDs and provides a liquid biopsy alternative to current diagnostic methods.

## 1. Introduction

Neurodegenerative disorders (NDDs) are a growing concern in the field of geriatric medicine as they are becoming increasingly prevalent in the aging population [1]. These disorders are primarily characterized by the loss of neurons and are associated with a diverse array of pathophysiological mechanisms that include memory impairment, cognitive decline, movement disorders, and other debilitating symptoms [2]. The most common NDDs are Alzheimer’s disease (AD) and Parkinson’s disease (PD). AD is a progressive disorder that leads to irreversible loss of memory and cognitive function. It is characterized by the presence of intracellular neurofibrillary tangles and senile plaques, which are caused by the accumulation of hyperphosphorylated microtubule-associated tau protein and amyloid-β (Aβ) peptides, respectively [3]. PD is the second most prevalent NDD, affecting approximately 2% of the population over the age of 60 [4]. The etiology of PD remains largely unknown, although a combination of genetic and environmental factors have been implicated in its pathogenesis [4]. Motor dysfunction in PD patients is due to the progressive loss of dopaminergic neurons in the substantia nigra pars compacta, and the presence of Lewy bodies, which are intracellular inclusions composed of α-synuclein [5].

A major public health challenge concerning NDDs is the lack of reliable and early diagnostic markers for these disorders. Alterations in protein aggregation, synaptic transmission, and mitochondrial pathways are common denominators in NDDs [6]. The identification of biomarkers that are accurate and accessible could significantly aid the early diagnosis of NDDs and the implementation of precision medicine programs. However, there are currently no reliable biomarkers for the diagnosis, classification, or for determining the progression of NDDs [7]. The most commonly utilized biomarkers are based on costly and/or invasive techniques such as neuroimaging or cerebrospinal fluid (CSF) analysis [8]. Neuroimaging techniques, such as amyloid positron emission tomography (PET) scanning with the novel tracers ^18^F-florbetaben, ^18^F-flutemetamol, and ^18^F-florbetapir are efficacious in the diagnosis of AD. Interestingly, the level of the major norepinephrine metabolite in the brain, 3-methoxy-4-hydroxyphenylglycol, in serum and CSF has been suggested as a new marker to differentiate AD from dementia with Lewy bodies and PD [9]. Liquid biopsy, a more cost- and time-effective and less invasive approach, has emerged as a promising option for the detection of biomarkers. In recent years, several lines of research have been focused on identifying new liquid biopsy biomarkers using different, more accessible, fluids such as urine, saliva, and blood. However, definitive liquid biopsy biomarkers for NDDs have yet to be identified. Furthermore, dysregulation of microRNA expression in peripheral blood has the potential for diagnosing AD and other NDDs. Moreover, data from a large-sample prospective study identified two blood biomarkers for VaD, asymmetric dimethylarginine (ADMA) and oxidized low-density lipoprotein (oxLDL) that appear to be promising [10]. The identification of plasma biomarkers for the diagnosis of NDDs has the potential to improve disease detection accuracy and specificity, particularly in separating AD from other clinical pathologies. Furthermore, the use of plasma biomarkers may improve the early diagnosis of AD, perhaps years before clinical symptoms appear. Several large, independent, cohorts of patients with AD showed consistent correlations between plasma Aβ42/Aβ40 levels and amyloid PET status [11]. Moreover, the plasma Aβ42/Aβ40 ratio, determined using an antibody-free mass-spectrometric (MS) approach, detects early pathogenic alterations in AD [12]. In addition to the emerging role of plasma Aβ as a biomarker for AD, plasma measurements of tau phosphorylated at threonine 231 (p-tau231), p-tau181 and p-tau217 have also shown diagnostic potential for the disease [13,14,15]. In fact, plasma p-tau217 levels in CSF exhibit changes that coincide with the appearance of amyloid plaques and precede tau-PET-positivity by 15 to 20 years [16], suggesting that plasma p-tau217 may be a useful tool for monitoring the pharmacodynamic effects of anti-amyloid drugs on amyloid pathology [17].

There is presently no cure or effective treatment for AD. In AD, neuronal death often begins more than a decade before the first symptoms manifest, implying that a diagnosis is often made when neuron loss is irreversible. Early diagnosis of NDDs may provide a longer therapeutic window for future therapies to slow or stop the neurodegeneration associated with AD, which could subsequently have a significant influence on patient survival and quality of life. Gene expression can be affected by a variety of factors, including genetic factors, exposure to toxins, and age. Comparisons of mRNA levels between healthy and diseased individuals allow for the identification of differentially-expressed genes, which may be causes, consequences, or mere correlates of the disease [18].

The aim of the current study was to explore novel biomarkers for the diagnosis of NDDs, with emphasis on the differential expression of several genes that are associated with neurodegeneration. Through the utilization of cutting-edge techniques, we have identified several genetic markers whose expression levels differ between healthy individuals and those with NDDs, suggesting that these markers, in the form of liquid biopsy biomarkers, may serve as valuable diagnostic tools for NDDs.

## 2. Results

### 2.1. Neurodegeneration-Related Gene Expression Is Altered in Patients with NDDs

NDD-related gene expression was analyzed in healthy individuals (*n* = 9), and in patients with AD (*n* = 7), PD and PD-related disorders (*n* = 14), and VaD/mixed dementia (*n* = 13) (Table 1). Apolipoprotein E (APOE) is a major risk factor for the development of AD [19]. We first compared *APOE* expression in buffy coat samples from healthy individuals with *APOE* expression in samples from patients with AD, PD, and PD-like disorders, and AD-like disorders such as vascular (VaD) and mixed dementia. *APOE* mRNA expression significantly decreased from 0.25 ± 0.014 in healthy individuals to 0.002 ± 0.001 in patients with AD (*p* < 0.05) (Figure 1A). There were no significant differences between healthy subjects and patients with PD or VaD/mixed dementia.

Mutations in the presenilin (*PSEN*) gene are known to cause early-onset familial AD [20]. Patients with AD had *PSEN1* mRNA levels of 0.34 ± 0.18, which were significantly lower than in healthy individuals (1.31 ± 0.33) (*p* < 0.05) (Figure 1B). There were no statistically significant differences in *PSEN1* mRNA expression between subjects with no-NDDs and patients with PD or VaD/mixed dementia.

Phosphatidylinositol-binding clathrin protein (*PICALM*) gene is a susceptibility locus for the incidence of late-onset AD (LOAD) [21]. We found no differences in the levels of *PICALM* expression between non-ND subjects and AD patients (Figure 1C). However, *PICALM* mRNA expression in patients with VaD/mixed dementia (29.1 ± 17.26) was 98% higher than in patients without NDDs (0.61 ± 0.3) (*p* < 0.05).

ATP-binding cassette, subfamily A, member 7 (*ABCA7*) is one of the most important risk factors for both early- and late-onset AD [22]. *ABCA7* mRNA levels decreased by 72% from 2.57 ± 0.7 in non-NDD subjects to 0.73 ± 0.36 in patients with AD (*p* < 0.05) (Figure 1D); no differences in *ABCA7* expression were detected between healthy subjects and patients with PD. However, *ABCA7* mRNA expression in patients with VaD/mixed dementia (2.57 ± 0.7) was 75% lower than in patients without NDDs (0.64 ± 0.24) (*p* < 0.05).

Alpha synuclein (SNCA) is a major component of Lewy bodies in PD [23]. Five mutations in the *SNCA* gene have been linked to autosomal dominant PD [23]. *SNCA* mRNA levels increased from 6.92 ± 3.35 in healthy subjects to 73,361.12 ± 52,458.3 in patients with PD (*p* < 0.05) (Figure 1G). There were no statistically significant differences in *SNCA* mRNA expression between individuals with no-NDDs and patients with AD or VaD/mixed dementia.

Opioid receptor kappa 1 (*OPRK1*) regulates cognitive and learning processes by inhibiting neurotransmitter release [24]. Moreover, there is a greater number of OPRK1 binding sites in the limbic system in AD brains than in healthy subjects [25]. There were no statistically significant differences in *OPRK1* mRNA levels between non-NDD subjects and patients with AD, PD, or VaD/mixed dementia (Figure 1E). The tyrosine receptor kinase (TRK) signaling pathway regulates neuronal development and plasticity, and there is a strong genetic link between neurotrophic receptor tyrosine kinase 2 (*NTRK2*) gene expression and AD [26]. We did not detect any differences in *NTRK2* mRNA expression between healthy individuals and patients with NDDs (AD, PD, or VaD/mixed dementia) (Figure 1F). Mutations in leucine-rich repeat kinase 2 gene (*LRRK2*) are linked to inherited and sporadic PD [27]. Similarly, we also found no differences in *LRRK2* mRNA levels between healthy subjects and patients with NDDs, including PD (Figure 1H).

To assess the diagnostic accuracy of *APOE*, *PSEN1*, *PICALM*, *ABCA7* and *SNCA*, we generated receiver operating characteristic (ROC) curves and calculated the area under the curve (AUC) for each gene in patients with AD, VaD/mixed dementia, and PD and PD-like disorders. *APOE* expression showed high diagnostic accuracy for AD with an AUC of 0.886 (100% sensitivity and 71.4% specificity, *p* < 0.001) (Figure 2A). In contrast, the AUCs for PD and VaD/mixed dementia were lower (0.636 and 0.7, respectively), with 45.5% sensitivity and 100% specificity for PD-like disorders (Figure 2B), and 70% sensitivity and 71.4% specificity for VaD/mixed dementia (Figure 2C).

ROC curve analysis of *PSEN1* expression also showed promising diagnostic accuracy for AD with an AUC of 0.852 (80.3% sensitivity, 77.8% specificity, *p* < 0.001) (Figure 2D). However, the diagnostic accuracy of *PSEN1* in PD-like disorders (Figure 2E) and VaD/mixed dementias (Figure 2F) was lower with an AUC of 0.681 (66.7% sensitivity, 77.8% specificity) and 0.713 (75% sensitivity, 66.7% specificity), respectively. The *p* value for *PSEN1* mRNA expression as a biomarker for patients with VaD/mixed dementia versus healthy subjects was 0.068, which was close to statistical significance.

ROC curve data in AD patients showed an AUC for *PICALM* of 0.55 (87.5% sensitivity, 40% specificity, *p* = 0.8) (Figure 2G); the AUC for *PICALM* in patients with PD and PD-like disorders was 0.569 (62.5% sensitivity, 66.7% specificity, *p* = 0.769) (Figure 2H). However, ROC curve analysis of *PICALM* expression revealed an AUC of 0.764 (44.4% sensitivity, 100% specificity; *p* = 0.028) in patients with VaD/mixed dementia (Figure 2I).

Analysis of the ROC curve for *ABCA7* mRNA expression as a biomarker for AD and AD-like disorders showed AUC values of 0.833 (66.7% sensitivity, 100% specificity, *p* = 0.006) (Figure 2J) and 0.828 (75% sensitivity, 100% specificity, *p* < 0.001) (Figure 2L). In samples from PD patients, the AUC was 0.688 (50% sensitivity, 100% specificity, *p* = 0.163) (Figure 2K).

The AUC from the ROC curve for *SNCA* mRNA expression in samples from AD patients was 0.667 (50% sensitivity, 100% specificity, *p* = 0.655) (Figure 2E). However, ROC curve data of *SNCA* mRNA levels as a biomarker for PD (Figure 2N) showed AUCs of 1 (100% specificity, 100% sensitivity, *p* < 0.001). The AUC from the ROC curve for VaD/mixed dementia was 0.7 (70% specificity and 71.4% sensitivity).

### 2.2. The ApoE4 Allele Alters APOE Expression Levels

The *E4* allele of the *APOE* gene is a strong risk factor for developing AD [28]; the *E2* allele, however, is protective against AD. To investigate the effect of the *APOE* gene on the levels of *APOE* expression, we next examined *APOE* expression in patients with different *APOE* genotypes (Table 2). Our patient sample included individuals with the following *APOE* genotypes: *APOE 2.3*, *APOE 3.3*, *APOE 2.4*, *APOE 3.4*, and *APOE 4.4.* Due to limited sample availability, we were only able to obtain samples from healthy individuals with the *APOE 3/3* and *APOE 3/4* genotypes. Compared to patients with the *APOE 3.3* genotype, *APOE* mRNA expression decreased in patients with NDDS: by 64% in subjects with the *APOE 2.4* genotype, 51% in those with the *APOE 3.4* genotype, and 73% in patients with the *APOE 4.4* genotype (Figure 3A). On the contrary, we did not find statistically significant differences between *APOE 3.3* and *APOE 3.4* genotypes in healthy subjects (Figure 3B).

### 2.3. PSEN1, PICALM, ABCA7 and SNCA Genotypes Do Not Affect Their Gene Expression

Next, we wanted to determine if there were any variations in the levels of gene expression among individuals with different NDD-related gene polymorphisms. We analyzed gene expression levels in patients with polymorphisms in *PSEN1*, *PICALM*, *ABCB7* and *SNCA* (Appendix A). The polymorphism in intron 8 (rs165932) of *PSEN1* is associated with increased susceptibility to LOAD [29]. In the current study, there was a slow progressive induction in *PSEN1* expression in heterozygous (GT) and homozygous (TT) pathological polymorphisms compared to the non-pathological (GG) polymorphism; however, these changes were not statistically significant (Figure 4A). The rs3851179 polymorphism in the *PICALM* gene is a prominent locus that contributes to AD [30]. However, there were no significant differences in the levels of *PICALM* expression among the different genotypes (Figure 4B). In terms of the *ABCA7* gene, the rs3764650 polymorphism is a susceptibility locus for AD [31,32]. We observed a small decrease in *ABCA7* mRNA levels in patients with the *GG* genotype, but this was not statistically significant (Figure 4C). The rs356182 polymorphism in the SNCA gene is related to tremor and is a risk factor for PD [33]. There were no statistically significant differences in the levels of *SNCA* expression among the different genotypes with this polymorphism (Figure 4D).

## 3. Discussion

The three most prevalent forms of dementia are AD, VaD, and mixed dementia. The presence of lacunar infarcts and white matter lesions in patients with AD indicates a strong association between cardiovascular disease and AD. Furthermore, classic AD-associated pathological changes such as amyloid plaques and neurofibrillary tangles are found in elderly patients with VaD. The brain lesions associated with VaD and AD often co-occur and interact, which significantly increases the likelihood of a substantial decline in cognitive function [34]. To this day, The lack of effective diagnostic biomarkers for NDDs remains a major challenge in clinical practice. The aim of the current study was to discover new biomarkers for NDD diagnosis by identifying differences in neurodegeneration-related gene expression. The NDD patient cohort included subjects with dementia (AD- and AD-like-disorders such as mixed and vascular dementia), and patients with PD and PD-like disorders. We used buffy coat samples obtained from healthy individuals and individuals with NDDs to measure changes in neurodegeneration-related gene expression. Our major finding was that gene expression analysis has diagnostic value for NDDs, offering a non-invasive (liquid biopsy) alternative to current diagnostic methods. Specifically, we identified several genes that showed differential expression in patients with AD, PD, and VaD/mixed dementia compared to healthy individuals. *APOE*, *PSEN1*, and *ABCA7* mRNA expression decreased in patients with AD, while *PICALM* mRNA levels increased in subjects with VaD/mixed dementia. Patients with PD and PD-related disorders showed increased *SNCA* mRNA levels. *APOE* mRNA expression had high diagnostic accuracy for AD, and moderate accuracy for PD and VaD/mixed dementia, while *PSEN1* mRNA expression showed promising accuracy for AD. *ABCA7* and *SNCA* mRNA expression showed high-to-excellent diagnostic accuracy for AD and PD, and moderate-to-high accuracy for VaD/mixed dementia. The *APOE E4* allele reduced *APOE* expression in patients with different APOE genotypes, but there was no association between *PSEN1*, *PICALM*, *ABCA7*, and *SNCA* gene polymorphisms and expression.

APOE is the major lipid and cholesterol carrier in the CNS [35]; it is important in lipoprotein metabolism in the brain and in the periphery, and is implicated in dementia and in ischemic heart disease [36]. In the current study, *APOE* mRNA levels decreased in venous blood samples from AD patients, suggesting that it may be a useful biomarker for the diagnosis of AD. More specifically, we also performed ROC curve analysis for five genes (*APOE*, *PSEN1*, *PICALM*, *ABCA7*, and *SNCA*) that are known to be regulated in NDDs. ROC curve analysis is a widely used tool for evaluating the diagnostic performance of biomarkers or diagnostic tests. By offering a comprehensive overview of sensitivity trends across all cutoffs, ROC curve analysis provides insights into the correlation between biomarker sensitivity and specificity [37,38]. ROC curves are particularly useful because they allow the calculation of the AUC, which represents the overall performance of the biomarker. In the current study, ROC curve analysis was used to assess the diagnosis accuracy of *APOE*, *PSEN1*, *PICALM*, *ABCB7* and *SNCA* mRNA levels in distinguishing between different NDDs (AD, PD, and VaD/mixed dementia). The importance of the ROC curve analysis to our conclusions is that it allowed the quantitative determination of the accuracy of the identified biomarkers in distinguishing between these NDDs in terms of sensitivity and specificity. For example, *APOE* expression showed high diagnostic accuracy for AD with an AUC of 0.886, while *PSEN1* expression had an AUC of 0.852 for AD. These results suggest that *APOE* and *PSEN1* mRNA levels could be potential biomarkers for the early detection of AD. Similarly, *SNCA* mRNA levels had an AUC of 1 in distinguishing between PD and other NDDs, indicating that *SNCA* mRNA expression may be a specific biomarker for PD diagnosis. The identified biomarkers in the current study have the potential to provide more objective and accurate diagnoses, which could lead to earlier detection of NDDs and more personalized treatment programs. Our study therefore highlights the importance of identifying reliable and accurate diagnostic biomarkers for NDDs.

As previously mentioned, the ROC curve for *APOE* showed an AUC of 0.886 with 71.4% specificity and 100% sensitivity, indicating its strong potential as a biomarker for AD. This finding is consistent with other studies that link *APOE* levels in CSF or plasma with an increased risk of developing AD [39,40]. APOE methylation in peripheral blood has also been proposed as a diagnostic biomarker for AD [41]. There is no association between *APOE* expression and cognitive impairment after stroke [42], which concurs with our current findings that *APOE* mRNA levels are not regulated in patients with VaD/mixed dementia.

There are three important *APOE* gene polymorphisms in humans, *APOE2*, *APOE3*, and *APOE4*; of these, the *E4* allele is a major genetic risk factor for the development of LOAD [35]. *E4* carriers have low levels of APOE, which may contribute directly to AD progression [43]. Furthermore, the *E4* allele of APOE affects Aβ protein deposition and clearance and is associated with increased risk in individuals with sporadic AD [19]. For several decades, the prevalent belief about the origin of AD is that the accumulation of Aβ plaques and hyperphosphorylated tau tangles lead to neurodegeneration and cognitive impairment. However, the etiology of sporadic AD, which accounts for more than 95% of all cases of AD, remains unknown [44]. In dementia with a vascular component, there is compelling evidence that neurovascular dysfunction precedes Aβ accumulation, thus contributing to the progression and/or development of AD. Moreover, microhemorrhages/microinfarcts may be early contributors to AD through deterioration of the blood–brain barrier and reduction of cerebral blood flow, which impede Aβ clearance [45]. In the current study, the level of *APOE* mRNA expression was lower in *E4* carriers than in patients harboring the *APOE2.3* and *3.3* genotypes.

Genetic mutations that are directly linked to AD include those in the presenilin 1 (*PSEN1*) and *PSEN2* genes. Mutations in *PSEN1* are the most common cause of AD; the second most prevalent cause of AD is mutations in the *APP* gene. Mutations in the *PSEN2* gene that lead to AD, however, are uncommon [46]. *PSEN1* encodes the major component of γ secretase, which is responsible for the proteolytic cleavage of APP and NOTCH receptor proteins, and the subsequent formation of Aβ peptides [47]. PSEN1 protein levels are reduced in the hippocampus and cerebral cortex of patients with AD [48]. *PSEN1* mRNA expression levels are similar in brain and leukocytes in AD patients [49]. PSEN1 protein levels are lower in platelets (but not in leukocytes) from AD patients than in healthy controls [50]. In the present study, our data reveal that *PSEN1* mRNA levels are lower in AD patients than in healthy subjects. Although *PSEN1* transcript levels increase after ischemia [51], there were no changes in the expression of *PSEN1* mRNA in patients with VaD/mixed dementia.

*PICALM* is one of the most important susceptibility genes for LOAD [52,53]. There are significant associations between several SNPs in the *PICALM* locus with AD-related phenotypes such as the age of onset, hippocampal atrophy, cognitive functions, and tau or Aβ levels in CSF [30]. Numerous and independent GWAS have identified several SNPs in the *PICALM* gene that are strongly associated with LOAD, the most significant being rs3851179 [54]. Compared to subjects with no NDDs, *PICALM* mRNA levels are high in the blood and brain in patients with AD and are downregulated in patients with PD [55]. Moreover, PICALM proteins have significantly stronger associations with cognition than *PICALM* mRNA; the levels of *PICALM* transcripts are not regulated in AD pathology [56]. Our current findings revealed no significant differences in *PICALM* mRNA levels between healthy subjects and patients with AD. However, *PICALM* expression levels were significantly higher in patients with VaD than in individuals from either of those two groups. Genetic variation in *PICALM* is associated with VaD independently of the *APOE* genotype [57], and *PICALM* overexpression causes *PICALM* to have a dominant-negative effect [58]. Thus, both an overexpression and a reduction of *PICALM* can produce a similar phenotype [59]. *PICALM* overexpression in HEK293 cells increases the expression of genes that are associated with cholesterol metabolism [59]. This indicates that there is a consistent link between abnormal cholesterol metabolism and VaD [60,61].

Mutations in genes that encode transporter proteins affect the pathogenesis and treatment of brain diseases, and the *ABC* gene family is particularly important in AD [62]. Examination of methylation and blood gene expression of the *ABCA7* gene as a biomarker of AD shows that *ABCA7* mRNA expression is higher in AD patients than in healthy controls; in patients with AD, high *ABCA7* transcript levels are linked to disease progression and a decline in cognitive function [63]. However, in our study, we found the opposite showing a significant decrease in *ABCA7* expression in AD patients. It has also been shown that having the protective rs3764659 (T) allele is linked to increased *ABCA7* expression [64].

PD is a form of multisystemic α-synucleinopathy characterized by the presence of Lewy bodies in the midbrain. Elevated expression of wild-type *SNCA* causes early-onset familial PD [65]. The normalization of overexpressed α-synuclein with moderate gene silencing RNA interference (expression control RNAi, ExCont-RNAi) substantially improves motor function [66]. In patients with Lewy body disease (LBD), *SNCA* transcript levels are increased in the brain but are decreased in the blood [67]. *SNCA* mRNA expression is reduced in early LBD and increases in early PD, suggesting that it may serve as a biomarker for the diagnosis of early LBD [67]. Our findings showed an increase in *SNCA* mRNA levels in PD and PD-related disorders. Furthermore, the *SNCA* rs356182 variant within the intercluster region increases *SNCA* expression in the human frontal cortex [68,69]. There were no significant differences in *SNCA* expression levels between the different genotypes with this polymorphism in the present study.

The fundamental objective of NDD biomarker research is to provide clinicians with neuroimaging or biochemical techniques that will aid in the diagnosis and surveillance of NDD activity. The field of preclinical diagnosis of AD through the use of liquid biopsy has undergone considerable advancements in recent years [70]. Given the limitations of the current biomarkers in accurately differentiating between various forms of dementia [71], the development of new liquid biopsy biomarkers is therefore important for the diagnosis and differentiation of NDDs such as AD and VaD, which often present with several similar symptoms. Indeed, the current biomarkers have an important limitation in their ability to discern among the different types of dementia. A combination of using plasma tau phosphorylated at threonine 181 (p-tau-181), neurofilament light chain (Nfl), and glial fibrillary acidic protein (GFAP) for the discrimination of AD, frontotemporal dementia, and dementia with Lewy bodies, has been recommended [72]. Aβ42 and Aβ40 are among the most extensively researched blood markers for diagnosing AD, and high plasma levels of Aβ42 correlate with the presence of the disease [73]. Tau phosphorylated at threonine 181 (p-tau-181) in blood is a highly specific and easily-accessible biomarker for diagnosing AD [74]. Plasma tau, when measured using ultrasensitive assays, is highly elevated in patients with dementia, but not to the same extent as in CSF [75]. However, there is no distinguishable difference between patients with mild cognitive impairment (MCI) who eventually develop AD and those with stable MCI [70].

In the present study, we propose that analyzing *APOE*, *PSEN1*, *PICALM*, *ABCB7*, and *SNCA* gene expression can be used to differentiate between PD, AD, and VaD/mixed dementia (Table 3). *APOE* and *PSEN1* mRNA levels are downregulated in blood samples from AD patients, *SNCA* expression is upregulated in patients with PD, *PICALM* expression is upregulated in patients with VaD/mixed dementia, and *ABCB7* transcript levels are downregulated in subjects with AD and in VaD/mixed dementia. Recent studies have also proposed global DNA methylation as a diagnostic and/or prognostic biomarker for AD, PD, and VaD [76,77]. Since sirtuin activity, brain-derived neurotrophic factor (BDNF) expression, and global DNA methylation levels significantly decrease in patients with PD or dementia, our group previously proposed the integration of these three epibiomarkers to improve NDD diagnostic accuracy [77]. However, these and other biomarkers have yet to be incorporated into routine clinical practice.

## 4. Materials and Methods

### 4.1. Subjects

For this retrospective study, patients were recruited from the CIBE Database at EuroEspes International Center of Neuroscience and Genomic Medicine (C000925, 21 October 2013, EuroEspes Biomedical Research Center). The study was conducted in accordance with the Helsinki Declaration, Spanish law (Organic Law on Biomedical Research, 14 July 2007), and following the approval of the Ethics/Research Committee of the EuroEspes Biomedical Research Center (Epibiomarkers EE0620). The samples were collected after the informed consent of all patients and/or legal caregivers. Following a comprehensive genetic and clinical examination, patients were diagnosed using globally-accepted diagnostic criteria. Patients with NDDs included those with dementia (AD and AD-like disorders such as vascular (VaD) and mixed dementia), and PD and PD-like disorders. Genomic analysis of many several single nucleotide polymorphisms (SNPs) associated with AD, PD, or vascular risk/mixed dementia, as well as psychological tests, brain mapping, and neuroimaging were included in the clinical protocol for patients. In the present study, the Mini-Mental State Examination (MMSE), a widely recognized and commonly used screening tool for evaluating cognitive impairment, was administered to all patients.

### 4.2. Sample Collection and Analysis

Venous blood samples were collected from individuals in the supine position following overnight fasting. EDTA-coated tubes from peripheral blood were centrifuged at 3000 rpm for 10 min at 4 °C and the buffy coat was then collected and stored at −40 °C until DNA extraction.

### 4.3. RNA Extraction

RNA was isolated from peripheral blood lymphocytes using the miRNeasy Mini Kit (Qiagen, Hilden, Germany) according to the manufacturer’s instructions. Samples were incubated at room temperature for 5 min before being mixed with chloroform, and were then centrifuged at 12,000× *g* for 15 min at 4 °C to separate the upper aqueous and organic phases. RNA was extracted using the QIAcube according to the manufacturer’s protocol. A microplate reader (Epoch, BioTek, Winooski, Vermont, USA) was used to determine RNA concentrations and quality. Only RNA samples with 260/280 and 260/230 ratios above 1.8 were used.

### 4.4. RT-qPCR

The high-capacity cDNA reverse transcription kit (Applied Biosystems, Foster City, CA, USA) was used to reverse transcribe RNA. Purified RNAs (200 ng) were copied into cDNAs using gene-specific primers with the following thermocycling conditions: 25 °C for 10 min, 37 °C for 120 min, and 85 °C for 5 min.

Gene expression was quantified by quantitative real-time RT-PCR (qPCR) using the StepOne Plus Real-Time PCR system (Applied Biosystems, Foster City, CA, USA). Each PCR reaction was carried out in duplicate, using the TaqMan Gene Expression Master Mix (Thermo Fisher, Waltham, MA, USA) and specific TaqMan probes (Thermo Fisher, Waltham, MA, USA) (Table 4). Results were then normalized to human glyceraldehyde 3-phosphate dehydrogenase (GAPDH) as an endogenous reference gene. Data analysis was done using the comparative CT method with the StepOne Plus Real-Time PCR software, and are presented as mean ± S.E.M.

### 4.5. Genotyping

We sought to investigate the effects that polymorphisms and genetic variation in different NDD-related genes have on their mRNA expression levels. SNPs and copy number variants (CNVs) were genotyped by qPCR with Taqman assays using the Step one Plus Real Time PCR System (Life Technology, Waltham, MA, USA) and Taqman OpenArray DNA microchips for the QuantStudioTM 12K Flex Real-Time PCR System. Genotyper software (Thermo Fisher Scientific, Waltham, MA, USA) was used to analyze the genotyping data.

### 4.6. Statistical Analysis

Data were tested for normality and equality of variances with the D’Agostini-Pearson and Levene’s tests, respectively. Statistical significance was determined with a one-way ANOVA with Bonferroni post hoc comparisons (GraphPad Prism, CA). Receiver operating characteristic (ROC curves) were plotted using MedCalc software (version 16.4.3; Ostend, Belgium). For ROC curves, the Youden’s index J (= maximum {sensitivity + specificity − 1}) was used to determine the optimum biomarker cut-off values, indicating the highest sensitivity and specificity for each marker. The area under the curve (AUC) and the associated *p*-values for each ROC curve were reported using Delong’s test; 95% confidence intervals of the AUCs were estimated with the exact binomial method. The area under the ROC curve provides a number between 0 and 1, where values from 0.9–1 are considered excellent, 0.8–0.9 are very good, 0.7–0.8 good, 0.6–0.7 sufficient, 0.5–0.6 bad and <0.5 not useful [78]. Higher AUC values correlate to better biomarker diagnostic power. Data are presented as mean ± S.E.M.; * *p* < 0.05, ** *p* < 0.01, and *** *p* < 0.001 were considered statistically significant.

## 5. Limitations of the Study

The findings of the current study will need to be confirmed using larger sample sizes. Furthermore, while our study only used buffy coat samples obtained from peripheral blood, it would be important to validate the findings of this study by using, for example, cerebrospinal fluid to reflect changes in gene expression that occur in the brain.

To investigate the effect of the *APOE* gene on *APOE* expression levels, we analyzed *APOE* mRNA levels in patients with different APOE genotypes. While including data on *APOE* gene expression in healthy individuals for comparison purposes would be valuable, we were unable to obtain samples from healthy subjects for each genotype. Nevertheless, we obtained samples from healthy individuals with two of the genotypes (*APOE 3/3* and *APOE 3/4*) in our subject group. Future studies incorporating all three genotypes (*APOE 2/3*, *APOE 2/4*, and *APOE 4/4*) would provide a more comprehensive understanding of the role of APOE genotypes in *APOE* expression, and would aid in the validation of potential targets for therapeutic interventions in AD. Finally, data normalization using alternative housekeeping genes (e.g., *RPL13A*) with lower biological variation should be considered for qPCR analyses to address the potential variability with regard to commonly used housekeeping genes such as *GAPDH* and β-tubulin (*β-TUB*).

## 6. Conclusions

In this study, we analyzed gene expression levels in healthy individuals and in patients with NDDs to identify novel biomarkers for NDD diagnosis. Our findings revealed that the analysis of *APOE*, *PSEN1*, *PICALM*, *ABCB7* and *SNCA* mRNA levels can be used to differentiate between PD, AD, and VaD/mixed dementia, and may serve as potential biomarkers for the early detection of these NDDs. Our study further highlights the impact of *APOE* gene variants in NDD patients, where *E4* carriers exhibit low *APOE* mRNA levels, which may directly contribute to the development of AD. Our study suggests that gene expression analysis may be a liquid biopsy alternative to the current diagnostic methods, thus addressing the lack of effective diagnostic biomarkers for NDDs. Moreover, the identified biomarkers may be clinically useful for the early identification of NDDs, which could help facilitate the development and implementation of a personalized treatment program for these diseases.

## Figures and Tables

**Figure 1 ijms-24-05746-f001:**
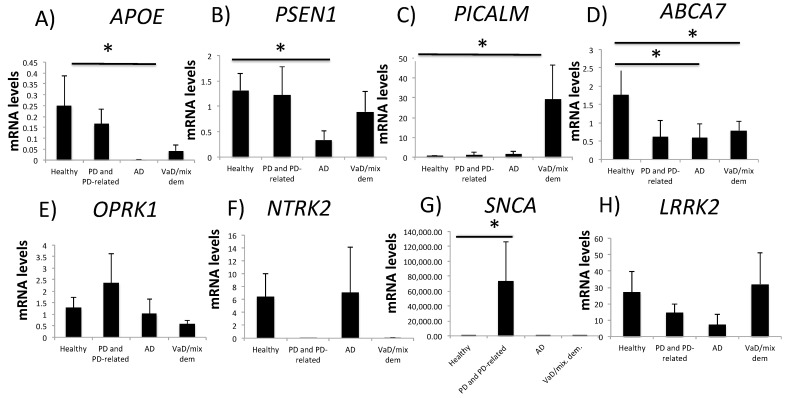
Gene expression in blood (buffy coat) samples from healthy subjects and patients with NDDs. Gene expression was analyzed in buffy coat samples from individuals with no-NDDs (*n* = 9) and patients with AD (*n* = 7), PD and PD-related disorders (*n* = 14), and VaD/mixed dementia (*n* = 13). qPCR with TaqMan probes for (**A**) *APOE*, (**B**) *PSEN1*, (**C**) *PICALM*, (**D**) *ABCA7*, (**E**) *OPRK1*, (**F**) *NTRK2*, (**G**) *SNCA* and (**H**) *LRRK2* were used. One-way ANOVA with post hoc Bonferroni corrections (* *p* < 0.05). Data are presented as means ± S.E.M. NDD, neurodegenerative disorders; AD, Alzheimer’s disease; ABCA7, ATP-binding cassette, subfamily A, member 7; APOE, apolipoprotein E; PD, Parkinson’s disease; PICALM, phosphatidylinositol-binding clathrin protein; PSEN1, presenilin-1; qPCR, quantitative real-time PCR; SNCA, alpha synuclein; VaD, vascular dementia.

**Figure 2 ijms-24-05746-f002:**
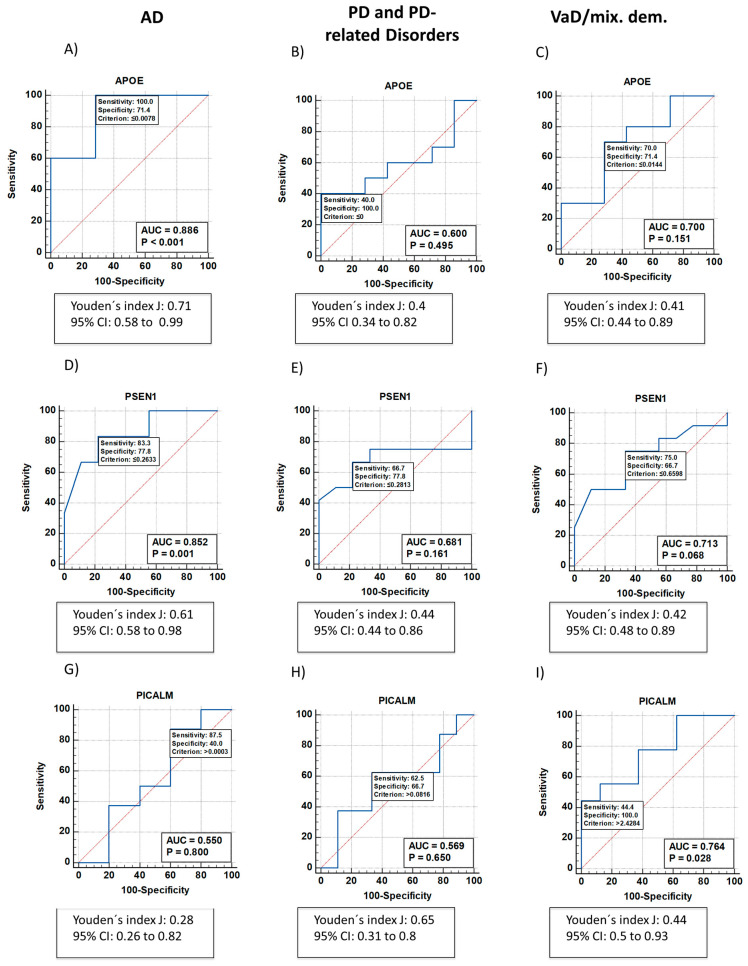
ROC curve analysis of mRNA expression data from buffy coat samples from patients with AD, PD and PD-like disorders, and VaD/mixed dementia. ROC curve analysis of *APOE* (**A**–**C**), *PSEN1* (**D**–**F**), *PICALM (***G**–**I**), *ABCA7* (**J**–**L**) and *SNCA* (**M**–**O**) mRNA expression from healthy (*n* = 9) subjects and patients with AD (*n* = 7; (**A**,**D**,**G**,**J**,**M**)), PD and PD-related disorders (*n* = 14; (**B**,**E**,**H**,**K**,**N**)), and VaD/mixed dementia (*n* = 13; (**C**,**F**,**I**,**L**,**O**)). The optimal cutoff values derived from the Youden’s J index are indicated below each graph. The red diagonal line is the ROC curve reference line. *p* values were calculated using Delong’s test; *p* < 0.05 was considered statistically significant. 95% confidence interval, 95% CI; AD, Alzheimer’s disease; ABCA7, ATP-binding cassette, subfamily A, member 7; APOE, apolipoprotein E; AUC, area under the curve; PD, PD, Parkinson’s disease; PICALM, phosphatidylinositol-binding clathrin protein; PSEN1, presenilin-1; ROC, receiver operating characteristic; SNCA, alpha synuclein; VaD, vascular dementia.

**Figure 3 ijms-24-05746-f003:**
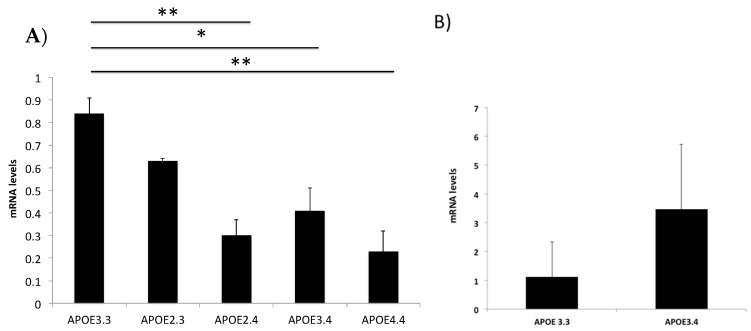
Gene expression in patients with different *APOE* genotypes. (**A**) *APOE* gene expression was analyzed in buffy coat samples from patients with the *APOE 3.3*, *APOE 2.3*, *APOE 2.4*, *APOE 3.4* and *APOE 4.4* genotypes, and in (**B**) healthy individuals with the *APOE 3.3* and *APOE 3.4* genotypes. Data are presented as means ± S.E.M. One-way ANOVA with Bonferroni corrections or t student (* *p* < 0.05; ** *p* < 0.01).

**Figure 4 ijms-24-05746-f004:**
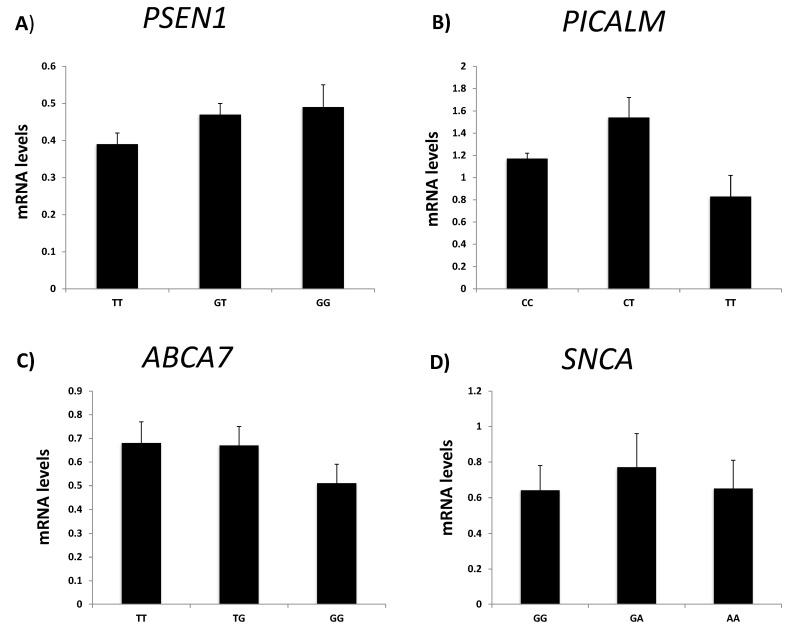
Gene expression in patients with different *PSEN1*, *PICALM*, *ABCA7* and *SNCA* genotypes. (**A**) *PSEN1*, (**B**) *PICALM*, (**C**) *ABCA7* and (**D**) *SNCA* mRNA levels were analyzed in buffy coat samples from patients with different genotypes. The labels for each bar on the *x*-axis refer to the different genotypes detected in the polymorphisms analyzed. Data are presented as means ± S.E.M. ABCA7, ATP-binding cassette, subfamily A, member 7; PICALM, phosphatidylinositol-binding clathrin protein; PSEN1, presenilin-1; SNCA, alpha synuclein.

**Table 1 ijms-24-05746-t001:** Patient demographics and clinical diagnosis.

Group	Clinical Diagnosis	MMSE	Age (Years)
Healthy	Healthy (*n* = 9)	29.18 ± 1.18	61.36 ± 8.58
AD	AD (*n* = 8)	14.92 ± 4.94	69.76 ± 7.9
VaD/mixed dementia	Vascular encephalopathy multi-infarction Binswanger-like (*n* = 4)	18.2 ± 4.8	77.6 ± 5.84
	Stroke (*n* = 2)		
	Mixed Dementia (Vascular-hypovitaminosis)		
	Vascular encephalopathy (*n* = 2)		
	Ischemic vascular encephalopathy (*n* = 4)		
PD and PD-related disorders	Parkinson’s (*n* = 7)	26.4 ± 2.26	67.8 ± 8.15
	Left vascular hemiparkinsonism		
	Incipient Parkinsonism (*n* = 2)		
	Parkinson’s with VaD (*n* = 2)		
	Familial Parkinson’s		

AD, Alzheimer’s disease; MMSE, Mini-Mental State Examination; VaD, vascular dementia.

**Table 2 ijms-24-05746-t002:** Demographics and diagnosis of patients with different *APOE* genotypes.

GENOTYPE	DIAGNOSIS	AGE (Years)	SEX
*APOE 3.3*	NCD mixed dementia	80	F
	PD	65	F
	NCD mixed dementia	80	M
	NCD mixed dementia	71	F
	Healthy (*n* = 4)	68 ± 8.12	2M, 2F
*APOE 2.3*	PD	69	F
	NCD	65	M
	BIPOLAR DISORDER	64	M
	NCD	61	M
*APOE 2.4*	NCD mixed dementia	78	M
	PD	75	F
	Depression	74	F
	Maturational delay	22	M
	NCD	75	M
*APOE 3.4*	NCD	76	M
	NCD	71	F
	NCD mixed dementia	65	M
	Depression	40	F
	Stroke	60	M
	Healthy (*n* = 4)	66.25 ± 12.5	2M, 2F
*APOE 4.4*	Encephalopathy	63	M
	NCD	69	F
	NCD	68	M
	NCD-AD	70	F
	NCD	70	M

AD, Alzheimer’s disease; APOE, Apolipoprotein E; NCD, neurocognitive disorder; PD, Parkinson’s disease; Sex: M, male; F, female.

**Table 3 ijms-24-05746-t003:** Summary of gene expression data.

	*APOE*	*PICALM*	*OPRK1*	*LRRK2*	*ABCA7*	*PSEN1*	*NTRK2*	*SNCA*
AD	low	-	-	-	low	low	-	-
VaD/mixed dementia	-	high	-	-	low	-	-	-
PD and PD-related disorders	-	-	-	-	-		-	high

*ABCA7*, ATP-binding cassette, subfamily A, member 7; *SNCA*, alpha synuclein; AD, Alzheimer’s disease; *APOE*, apolipoprotein E; *LRRK2*, leucine-rich repeat kinase 2; *NTRK2*, neurotrophic receptor tyrosine kinase 2; *OPRK1*, opioid receptor kappa 1; PD, Parkinson’s disease; *PICALM*, phosphatidylinositol binding clathrin assembly protein; *PSEN1*, presenilin-1; VaD, vascular dementia.

**Table 4 ijms-24-05746-t004:** TaqMan probes.

GENE	ID
*APOE*	Hs00171160_m1
*PSEN1*	Hs00240518_m1
*PICALM*	Hs00300318_m1
*ABCA7*	Hs1105094_m1
*OPRK1*	Hs00175127_m1
*NTRK2*	Hs00178811_m1
*SNCA*	Hs00240906_m1
*LRRK2*	Hs00968192_m1

## Data Availability

Not applicable.

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
