# Peer review of "Gene Expression Profiling as a Novel Diagnostic Tool for Neurodegenerative Disorders"

_ijms, 2023, doi:10.3390/ijms24065746_

Round 1

Reviewer 1 Report

In this manuscript, Martines-Igleaias and colleagues explored how gene expression profiles differ in different neurodegenerative disorder (NDD) patients. They tried to establish gene expression profiles for diagnosing Alzheimer's disease (AD), Parkinson's disease (PD), and vascular (VaD)/mixed dementia. Based on their results, the authors suggest that gene expression analysis has diagnostic value for NDDs, providing a liquid biopsy alternative to current diagnostic methods.

This is an interesting study that increases our understanding of novel biomarkers for the diagnosis of NDDs, with emphasis on the differential expression of several genes that are associated with neurodegeneration. However, many aspects of the results are not adequately addressed in the result and discussion parts. Especially, although this study provided an interesting perspective by comparing different gene expression profiles between healthy control and patients, there are a number of issues that should be better explained and organized. More comments are below.

1.    Several recent research papers investigated biomarkers for diagnosing NDDs using blood plasma samples, Ex) PMID: 34906975. It should be discussed in the introduction part.

2.    My main concern is that there is no patient biography information. The severity of NDDs in each patient must affect the gene expression profiles. Therefore, all subjects’ biography information should be addressed.

3.    Why were ROC curve analyses used for the main conclusions? The authors should describe the importance of this method for their conclusions.

4.    In figure 4, the labels for the x-axis should be edited, What is AA, GA, or CC for?

5.     Discussion part should be better organized to have a good connection to address the main and important findings in their research.

Author Response

Dear Ms. Zhang,

Thank you for giving us the opportunity to submit a revised draft of our manuscript titled “Gene Expression Profiling As A Novel Diagnostic Tool For Neurodegenerative Disorders”. We are grateful for the helpful criticism given by the reviewers and hope that the revised manuscript will be acceptable for publication as a Research Article in the special issue New Trends in Alzheimer's Disease Research: From Molecular Mechanisms to Therapeutics in International Journal of Molecular Sciences.

We have incorporated changes that reflect the suggestions provided by the reviewers and have highlighted those changes in blue in the manuscript. The points raised by the reviewers have been addressed as follows (reviewers’ comments are italicized).

REVIEWER 1

Comment 1:

Several recent research papers investigated biomarkers for diagnosing NDDs using blood plasma samples, Ex) PMID: 34906975. It should be discussed in the introduction part.

Response:

Thank you to the reviewer. We have added information concerning the diagnosis of NDDs using blood plasma samples, to the Introduction section of the revised manuscript.

Comment 2:

My main concern is that there is no patient biography information. The severity of NDDs in each patient must affect the gene expression profiles. Therefore, all subjects’ biography information should be addressed.

Response:

The authors apologize for this omission and thank the reviewer for pointing this out. We have now included, in the revised manuscript, a new Table 1 (Patient Demographics and their Clinical Diagnoses), Table 2 (Demographics and Diagnosis of Patients with Different APOE Genotypes), and Supplementary Table S1 (Supplementary Table S1. Diagnosis and Demographics of Patients with Different PSEN, PICALM, ABCA7 and SNCA genotypes).

Comment 3

Why were ROC curve analyses used for the main conclusions? The authors should describe the importance of this method for their conclusions.

Response:

ROC curve analysis was used for our main conclusions because it is a commonly used method for assessing the diagnostic capacity of biomarkers or diagnostic tests. This approach offers an insight into the correlation between biomarker sensitivity and specificity. We have now described the importance of ROC curves in Paragraph 2 of the Discussion section in the revised manuscript.

Comment 4:

In figure 4, the labels for the x-axis should be edited, What is AA, GA, or CC for?

Response:

This has been rectified in the revised manuscript. The labels (AA, GA, CC, GT, GG and so forth) for each bar on the X-axis refer to the different genotypes detected in the polymorphisms that were analyzed. We have now added X-axis titles for each graph to indicate this.

Comment 5:

Discussion part should be better organized to have a good connection to address the main and important findings in their research.

Response:

Thank you to the reviewer. The Discussion section has been restructured, containing additional information that addresses the principal findings in our study. We have also added a new Table 4 to the revised manuscript, which summarizes the expression data of eight genes (APOE, PICALM, OPRK1, LRRK2, ABCA7, PSEN1, NTRK2, and SNCA) from our study. The table shows that analyzing the expression of these genes can differentiate between AD, PD and PD-related disorders, and vascular/mixed dementia.

Reviewer 2 Report

In the last years, the interest for novel biomarkers for the diagnosis of the main neurodegenerative disorders has increased considerably. The manuscript by Martínez-Iglesias and co-workers is inserted in this vein. It is aimed at identifying novel blood biomarkers of Alzheimer, Parkinson, and Vascular dementia, by focusing attention of changes of mRNA levels of some genes linked to neurodegeneration.

Main concerns:

1) Subjects: more details about age, sex, and stage of the neurodegenerative disease for each patient (and the eventual genetic data) should be inserted in the paper (eventually as supplementary data).

2) In the Materials and Methods, it is mentioned that "Venous blood samples were collected from individuals in the supine position following overnight fasting."

Did patients undergo a wash-out period (and, eventually, for how long time)?

3) qPCR: normalization vs additional or alternative housekeeping gene/s with a lower biological variation than well known “housekeeping” genes like GAPDH and tubulin should be carried out.

4) Figure 3. Gene expression in patients with different APOE genotypes. The evaluation of APOE genotypes on APOE expression in healthy subjects is necessary for comparison purposes.

5) The addition of a table or figure summarizing the gene expression data would be helpful to visualize better the results and suggest a possible diagnostic algorithm.

Minor:

Correct LLRK2 -> LRRK2

Author Response

Dear Ms. Zhang,

Thank you for giving us the opportunity to submit a revised draft of our manuscript titled “Gene Expression Profiling As A Novel Diagnostic Tool For Neurodegenerative Disorders”. We are grateful for the helpful criticism given by the reviewers and hope that the revised manuscript will be acceptable for publication as a Research Article in the special issue New Trends in Alzheimer's Disease Research: From Molecular Mechanisms to Therapeutics in International Journal of Molecular Sciences.

We have incorporated changes that reflect the suggestions provided by the reviewers and have highlighted those changes in blue in the manuscript. The points raised by the reviewers have been addressed as follows (reviewers’ comments are italicized).

REVIEWER 2

Comment 1:

1) Subjects: more details about age, sex, and stage of the neurodegenerative disease for each patient (and the eventual genetic data) should be inserted in the paper (eventually as supplementary data).

Response:

Thank you to the reviewer. The authors apologize for this omission We have now included, in the revised manuscript, a new Table 1 (Patient Demographics and their Clinical Diagnoses) and Table 2 (Demographics and Diagnosis of Patients with Different APOE Genotypes). Supplementary Table S1 (Diagnosis and Demographics of Patients with Different PSEN, PICALM, ABCA7 and SNCA genotypes) has also been added to the submitted manuscript.

Comment 2:

In the Materials and Methods, it is mentioned that "Venous blood samples were collected from individuals in the supine position following overnight fasting". Did patients undergo a wash-out period (and, eventually, for how long time)?

Response:

The patients in our study did not undergo a wash-out period.

Comment 3:

qPCR: normalization vs additional or alternative housekeeping gene/s with a lower biological variation than well known “housekeeping” genes like GAPDH and tubulin should be carried out.

Response:

Thank you to the reviewer for this helpful comment. Unfortunately, given the limited (10-day) timeframe for manuscript resubmission, it is not possible for us to perform additional normalization with alternative housekeeping genes since we do not currently have the appropriate probes and materials in our laboratory, and would need to procure them. The GAPDH housekeeping gene was used a reference gene in our current study because of their predicted stable expression in line with previous studies by our group (please see citations below). Nonetheless, we appreciate this comment and agree with the reviewer that experiments using an additional housekeeping gene are important to perform. We have included this point as a limitation of the study in section 5 of the revised manuscript.

References:

1) Martínez-Iglesias, O., Naidoo, V., Cacabelos, N., & Cacabelos, R. (2021). Epigenetic Biomarkers as Diagnostic Tools for Neurodegenerative Disorders. International journal of molecular sciences, 23(1), 13.

2) Martínez-Iglesias, O., Carrera, I., Naidoo, V., & Cacabelos, R. (2022). AntiGan: An Epinutraceutical Bioproduct with Antitumor Properties in Cultured Cell Lines. Life (Basel, Switzerland), 12(1), 97.

3) Martínez-Iglesias, O., Carrera, I., Carril, J. C., Fernández-Novoa, L., Cacabelos, N., & Cacabelos, R. (2020). DNA Methylation in Neurodegenerative and Cerebrovascular Disorders. International journal of molecular sciences, 21(6), 2220.

Comment 4:

Figure 3. Gene expression in patients with different APOE genotypes. The evaluation of APOE genotypes on APOE expression in healthy subjects is necessary for comparison purposes.

Response:

Thank you to the reviewer for this insightful comment. We have now included in the revised manuscript a new Figure 3B. Although including data on APOE gene expression in healthy individuals for comparison purposes is important, we were unable to obtain samples from healthy subjects for each genotype due to limited sample availability. Nevertheless, we did obtain samples from healthy individuals with two of the genotypes (APOE 3/3 and APOE 3/4) in our subject group. We wish to acknowledge that future studies incorporating all three genotypes (APOE 2/3, APOE 2/4, and APOE 4/4) would provide a more comprehensive understanding of the role of APOE genotypes in APOE expression, and would aid in the validation of potential targets for therapeutic interventions in AD.

Comment 5:

The addition of a table or figure summarizing the gene expression data would be helpful to visualize better the results and suggest a possible diagnostic algorithm.

Response:

Thank you to the reviewer. A new Table 4 that summarizes the gene (APOE, PICALM, OPRK1, LRRK2, ABCA7, PSEN1, NTRK2, and SNCA) expression data from our study has been added to the revised manuscript. This table demonstrates that analyzing the expression of these genes can be used to differentiate between AD, PD and PD-related disorders, and vascular/mixed dementia.

Minor Comment:

Correct LLRK2 -> LRRK2

Response:

We apologize for this error. LLRK2 has been corrected to LRRK2 in the revised manuscript.

Reviewer 3 Report

This is a well-structured article. The main question addressed by this research is the fact that gene expression profiling can assist in diagnosing neurodegenerative diseases.

The introduction gives the neurobiological background of this study as it briefly describes the current knowledge about neurodegenerative diseases, the biomarkers that are used in their diagnosis and finally refers to the aim of the present study which is the potential use of differential expression of several genes that are associated with neurodegeneration as a diagnostic biomarker.

The results are quite interesting and, to my opinion, well presented.

The discussion is well written, summarizing and discussing the main findings of the study. Perhaps, a paragraph summarizing its main limitations would, to my opinion, add to the scientific value of the paper. Furthermore, the conclusions could be written in a separate section in order to be emphasized.

“Materials and Methods” section is descriptive enough. It refers to the subjects of the study, the samples’ collection and their analysis, RNA-extraction and genotyping techniques, as well as statistical analyses that were used during this study.

References are adequate in number and relative to the subject.

English language and style are generally fine.

Author Response

Dear Ms. Zhang,

Thank you for giving us the opportunity to submit a revised draft of our manuscript titled “Gene Expression Profiling As A Novel Diagnostic Tool For Neurodegenerative Disorders”. We are grateful for the helpful criticism given by the reviewers and hope that the revised manuscript will be acceptable for publication as a Research Article in the special issue New Trends in Alzheimer's Disease Research: From Molecular Mechanisms to Therapeutics in International Journal of Molecular Sciences.

We have incorporated changes that reflect the suggestions provided by the reviewers and have highlighted those changes in blue in the manuscript. The points raised by the reviewers have been addressed as follows (reviewers’ comments are italicized).

REVIEWER 3

Comment:

The discussion is well written, summarizing and discussing the main findings of the study. Perhaps, a paragraph summarizing its main limitations would, to my opinion, add to the scientific value of the paper. Furthermore, the conclusions could be written in a separate section in order to be emphasized.

Response:

Thank you to the reviewer. As suggested, we have included the “Limitations of the Study” (section 5) and a separate “Conclusions” section (6) in the revised manuscript.

Round 2

Reviewer 2 Report

The Authors have satisfactorily addressed most of my concerns and have significantly improved the manuscript.